

# Injury incidence, characteristics and burden among female sub-elite futsal players: a prospective study with three-year follow-up

Iñaki Ruiz-Pérez[1,*], Alejandro López-Valenciano[1,2,*],
Alejandro Jiménez-Loaisa[1], Jose L.L. Elvira[1], Mark De Ste Croix[3] and
Francisco Ayala[1,3]

[1] Department of Sport Sciences, Sports Research Centre, Miguel Hernández University, Elche, Alicante, Spain
[2] Universidad Internacional Isabel I de Castilla, Burgos, Spain
[3] School of Sport and Exercise, University of Gloucestershire, Gloucester, UK
* These authors contributed equally to this work.

Corresponding author
Francisco Ayala, fayala@umh.es

## ABSTRACT

The main purpose of the current study was to analyze the injury incidence, characteristics and burden among sub-elite female futsal players. Individual exposure to match play and training, injury incidence and characteristics (player position, injury mechanism, type of injuries, severity of injuries, recurrent vs. new injuries, season variation of injury pattern) in a female futsal team were prospectively recorded for three consecutive seasons (2015–2018). Incidences were calculated per 1,000 h of exposure. A total of 30 injuries were reported during the three seasons within a total exposure of 4,446.1 h. The overall, match and training incidence of injuries were 6.7, 6.4 and 6.8 injuries/1,000 h of exposure, respectively. Most injuries had a non-contact mechanism (93%), with the lower extremity being the most frequently injured anatomical region (5.62 injuries/1,000 h of exposure). The most common type of injury was muscle/tendon (4.9 injuries/1,000 h of exposure) followed by joint (non-bone) and ligament (1.3 injuries/1,000 h of exposure). The injuries with the highest injury burden were those that occurred at the knee (31.9 days loss/1,000 h exposure), followed by quadriceps (15.3 day loss/1,000 h) and hamstring (14.4 day loss/1,000 h) strains. The first few weeks of competition after pre-season and soon after the Christmas break were the time points when most injuries occurred. These data indicate that sub-elite female futsal players are exposed to a substantial risk of sustaining an injury. To reduce overall injury burden, efforts should be directed toward the design, implementation and assessment of preventative measures that target the most common diagnoses, namely, muscle/tendon and ligament injuries.

## INTRODUCTION

Futsal, the five-a-side version of associated football, is played worldwide with more than one million registered players all over the world (*FIFA, 2007*; *Gorostiaga et al., 2009*; *Beato*

*et al., 2017*). Futsal requires players to perform on a reduced (usually indoor) pitch size (40 × 20 m) and during 2 × 20 min periods (with time stopping at every dead ball and unlimited substitutions) a high number of repeated high intensity multiplanar movements such as sudden acceleration and deceleration, rapid changes of direction, tackling and kicking (*Castagna et al., 2009*; *Beato, Coratella & Schena, 2016*; *Naser, Ali & Macadam, 2017*). At top levels, the combination of these repeated high intensity movements that are performed during training and match play alongside current congested training and competitive calendars and exposure to contacts might place futsal players at high risk of injury. However, prior to implementing injury prevention programs into everyday futsal training routines, it is essential to establish the extent of the problem in terms of the incidence and characteristics of injuries (*Van Mechelen, Hlobil & Kemper, 1992*; *Finch, 2006*; *Van Tiggelen et al., 2008*).

Despite being one of the most played sport in several countries, a limited number of prospective epidemiological studies have been published investigating injuries sustained by elite futsal players (mainly during match play) (*Ribeiro, Oliveira & Costa, 2006*; *Junge & Dvorak, 2010*; *Angoorani et al., 2014*; *Hamid, Jaafar & Ali, 2014*; *Álvarez Medina et al., 2016*). These studies have reported incidence rates for male players ranging from 3.5 to 89.9 injuries per 1,000 h of match play, most of them affecting the lower extremity with contusions of the lower leg and ankle sprains the most frequently diagnosed types of injury (*Ribeiro, Oliveira & Costa, 2006*; *Junge & Dvorak, 2010*; *Angoorani et al., 2014*; *Hamid, Jaafar & Ali, 2014*; *Álvarez Medina et al., 2016*; *Larruskain et al., 2018*). However, it should be noted that among these epidemiological studies, only two (*Angoorani et al., 2014*; *Hamid, Jaafar & Ali, 2014*) have reported incidence data of female futsal players. *Angoorani et al. (2014)* showed an incidence rate in female players of 10.7 injuries per 1,000 h of match play during camps with the Iran national team (18 months of follow-up), whereas *Hamid, Jaafar & Ali (2014)* found an incidence rate of 19.7 injuries per 1,000 h of match play during the Malaysian national futsal league. In both studies, ankle sprains and ligament ruptures were the most observed injuries, similar to what has been observed in other team sports such as football (*Hägglund, Waldén & Ekstrand, 2009*; *Asker et al., 2018*), handball (*Asker et al., 2018*) and rugby (*Peck et al., 2013*). It is likely that the anatomical, hormonal and neuromuscular sex-related differences (among other factors) may contribute to sex-specific differences in injury incidence. Furthermore, only *Angoorani et al. (2014)* provided injury incidence rates during training in male and female futsal players, reporting an incidence of 1.8 and 3.1 injuries per 1,000 h of exposure, respectively. As the training volume (*Almeida et al., 1999*) and the number of hours of high intensity training (*Brooks et al., 2008*) have been significantly correlated with an increased risk of sustaining non-contact injuries in team sports (mainly attributed to an acute and/or cumulative fatigue state), knowing the injury incidence rates during futsal training may help coaches and physical trainers to identify if the training load and content allows players to recover fully from match demands. None of the studies that have provided epidemiological data of futsal-related injuries in male and female players have calculated the injury burden (the product of severity (consequences) and incidence (likelihood)) and/or built a risk matrix. A risk matrix is a graph of injury severity plotted against injury

incidence with criteria incorporated into the graph for evaluating the level of risk, usually by dividing the graph into some risk areas using descriptive or quantified incidence, severity and risk evaluation categories (*Fuller, 2018*).

Consequently, there is a clear need for more prospective epidemiological studies that inform about injury incidence and burden in female futsal players. Identifying the most common and burdensome futsal-related injuries, as well as how (traumatic or overuse) and when (matches or training sessions) they usually occur would lead coaches, physical trainers and physiotherapists to prioritize the application of specific measures to prevent or reduce the risk of sustaining such injuries. Therefore, the main purpose of the current study was to analyze the injury incidence, characteristics and burden among sub-elite female futsal players during three consecutive seasons.

## METHOD

### Participants

All female sub-elite futsal players from the same team that were playing in the Spanish second division were prospectively followed during three consecutive seasons (2015/16, 2016/17 and 2017/18) which covered the period between September and May. Twenty-two different female futsal players participated in this study. However, as some players remained in the team for more than one season, the total number of player seasons was 39 (2015/16: 14 players followed, 2017/17: 13 players followed, 2017/18: 12 players followed). All players had more than 5 years of futsal experience. The team finished all three seasons in the top 10 of the league (4st, 6st and 9st). All players were verbally informed about the study procedures and provided written informed consent. For players younger than 18 years old ($n = 3$), written informed consent was also obtained from their parent or legal guardian. Players who left the team during the season (e.g., due to transfer) were included in the analysis according to their time on the team. The experimental procedures used in this study were in accordance with the Declaration of Helsinki and were approved by the University Office for Research Ethics (*Órgano evaluador de proyectos, Universidad Miguel Hernández de Elche*) (DPS.FAR.02.14).

### Data collection

The study design and data collection followed both the consensus on definitions and data collection procedures for studies of football injuries outlined by the Union of European Football Associations (*Hägglund et al., 2005*) and the consensus document for football injury surveillance studies (*Fuller et al., 2006*). An injury was defined as any physical complaint sustained by a player that resulted from a futsal match or futsal training and where the player was unable to participate in a match or training sessions on the day after the injury (time-loss injury) (*Fuller et al., 2006*). The day on which an injury occurred was day 0 and was not counted when determining the severity of an injury. If a player had to stop training or participating in a match because of injury on 1 day but could participate the next day, the time loss was recorded as 0 days.

The club's medical staff (which remained the same for all three seasons), diagnosed, treated and recorded all time-loss injuries on a standardized injury report form that was

sent to the study group each month. Specifically, the team was supported by one certified medical doctor, one physical trainer and one physiotherapist. The doctor was the member of the medical staff who assessed and diagnosed injured players through the use of clinical judgements (e.g., physical examination, posture and gait inspection, inspection and palpation of muscle bellies, etc.). Diagnostic imaging techniques (e.g., echography, magnetic resonance imaging and ultrasound imaging) were also applied when it was needed. Although early treatment actions were delivered as soon as possible when a player sustained an injury during training or competition, the initial assessment and diagnosis were often carried out within 12 h to 4 days post-injury as some signs of injury may arise a few hours or days later (*Askling et al., 2007*). The physiotherapist administered the therapeutic exercises during the first stages of the rehabilitation process. The physical trainer was responsible for introducing injured players to the drills and skills that would be required to return to full participation in training and to be available for match selection. A futsal player was considered injured until the medical staff (upon agreement) allowed full participation in training and they were eligible for match play.

For all injuries that satisfied the inclusion criteria (time-loss injury), team medical staff provided the following details to investigators: date of injury, moment (training or competition), player position (goalkeeper or field player (lastwoman, wing or pivot)), injury mechanism (traumatic (contact or non-contact) or overuse), injury location, type of injury (the specific injury diagnosis was also recorded), extremity of the injury (dominant/non dominant), injury severity based on lay off time (0 days (when a player could not participate fully on the day of an injury but was available for full participation the next day), minimal (1–3 days), mild (4–7 days), moderate (8–28 days), severe (>28 days) and career ending injury), whether it was a recurrence or new injury and total time taken to resume full training and competition. Illnesses and any physical or mental complaint that did not result from a futsal match or training were excluded. Individual player exposure time in training and matches (friendly and competitive) were recorded daily in minutes by the physical trainer.

The operational definitions adopted by this study have been widely followed by both football and futsal epidemiological studies (*Hägglund, Waldén & Ekstrand, 2009*; *Junge & Dvorak, 2010*; *Ekstrand et al., 2013*; *Hamid, Jaafar & Ali, 2014*; *Larruskain et al., 2018*) and they are displayed in Appendix A1.

Those players who were already injured when the follow up process started (September 2015) were included in this study once medical staff agreed return to training and availability for match selection. Those individuals who were still injured at the end of the study period were included in the statistical analyses, and the estimated duration of the recovery period was established after discussion with the respective medical staff. As a medical history based on information from the player may be confounded by recall bias, previous injuries of those players who were recruited to the team after the study started were not included unless an accurate and detailed description of them were provided in the form of a report or standard form and signed by either a certified medical doctor or a former physiotherapist.

Demographic information such as stature, body mass and age were collected during the last week of the preseason period (which was before the start of the season).

## Data analysis

Descriptive data are presented as a mean with the corresponding standard deviation, proportions (%), incidence rates and 95% confidence intervals (CI). The overall injury incidence, match injury incidence and training injury incidence were the number of injuries divided by 1,000 player-hours in total, match and training, respectively. For incidence rates, 95% CIs were calculated as the incidence ±1.96 times the square root of the number of injuries divided by the number of participants. The injury burden was calculated as the number of lay-off days/1,000 h (*Bahr, Clarsen & Ekstrand, 2017*). Player overall hours were calculated by adding match and training hours. Player match hours were calculated by multiplying total number of matches in the season per five players per match duration (40 min with stopped clock)/60, and player training hours were calculated by adding individual training hours (warm up of the matches was not included). All of the analyses were performed using the PASW statistical package, version 18.0 (SPSS Inc., Chicago, IL, USA), with $p < 0.05$ considered statistically significant. A post hoc power analysis was conducted using the software package, G*Power 3.1.2 (*Faul et al., 2007*; *Faul et al., 2009*). The sample size of 39 was used for the statistical power analyses. The alpha level used for this analysis was $p < 0.05$. The post hoc analyses revealed the statistical power for this study was 0.74. It could be concluded that the given sample size was large enough to detect significant effects.

The spreadsheet designed by *Hopkins (2007)* for combining effect statistics was used to make clinically (qualitative) inference for paired-comparisons between incidence rates. In particular, the incidence rate ratio (and its associated confidence limits) was assessed against predetermined thresholds. Thus, an incidence rate ratio of 0.91 represented a substantially lower injury risk, while an incidence rate ratio of 1.10 indicated a substantially higher injury risk (*Hopkins, 2010*). An effect was considered unclear if its CI overlapped the thresholds just mentioned; in other words, if the effect could be substantial in both a positive and negative sense. Otherwise the effect was clear and deemed to have the magnitude of the largest observed likelihood value. The following scale was used to qualify with a probabilistic term the magnitude of the observed effect: <0.5%, most unlikely; 0.5–5%, very unlikely; 5–25%, unlikely; 25–75%, possible; 75–95%, likely; 95–99.5%, very likely; >99.5%, most likely (*Hopkins, 2007*).

## Study quality assessment

The quality of the study was assessed using the "Strengthening the reporting of observational studies in epidemiology" (STROBE) (*Von Elm et al., 2014*) and the risk of bias of external validity quality, using an adapted version of the Newcastle Ottawa Scale (NOS) (*Saragiotto et al., 2014*; *Videbæk et al., 2015*). The study fulfills all the criteria of the STROBE scale except the items 9 and 10 (Appendix A2). Regarding the NOS adapted scale just item 6 was not fulfilled (Appendix A3). Thus, the reporting and external validity
**Table 1 Players and team characteristics and exposure time.**

|  | Season 15/16 | Season 16/17 | Season 17/18 | Total | Mean |
|---|---|---|---|---|---|
| Team size | 14 (14) | 13 (12) | 12 (9) | 39 (35) | 13 ± 1 |
| Age (years) | 23.8 ± 2.9 | 24.2 ± 4.1 | 24.2 ± 4.8 | – | 24.1 ± 3.9 |
| Height (cm) | 1.65 ± 0.05 | 1.65 ± 0.04 | 1.65 ± 0.04 | – | 1.65 ± 0.04 |
| Body mass (kg) | 60.4 ± 5.1 | 62.3 ± 7.4 | 61.9 ± 7.4 | – | 61.5 ± 6.6 |
| Weeks of follow-up | 32 | 35 | 36 |  | 34.3 ± 2.1 |
| Exposure |  |  |  |  |  |
| Total h | 1506.7 | 1328.78 | 1610.7 | 4446.1 | 1482.1 ± 142.6 |
| Training h | 1413.3 | 1222.1 | 1500.7 | 4136.1 | 1378.8 ± 142.5 |
| Match h | 93.3 | 106.7 | 110 | 310 | 103.3 ± 8.8 |
| Training sessions/week | 4 ± 0.3 | 3.2 ± 0.7 | 3.8 ± 0.5 | – | 3.7 |
| Matches/week | 0.875 | 0.914 | 0.917 | – | 0.903 |
| Match exposure ratio[a] | 0.062 | 0.080 | 0.068 | – | 0.070 |
| Days of absence due to the injury | 234 | 144 | 51 | 429 | 143 ± 91.5 |

Notes:
  h, hours.
  Values are mean ± SD.
  [a] Match hours/total hours of exposure.

quality of the present study could be considered as high according to the qualitative descriptors proposed by *Von Elm et al. (2014)* and *Wells et al. (2013)* respectively.

# RESULTS

During the three seasons, four players dropped out due to transfers to another club or they were released by the club but their injury data were included based on their time at the club. The average duration of each season was 34.3 ± 2.1 weeks with 31 ± 2.7 matches per season and 3.3 ± 1.3 trainings sessions per week. Player and team characteristics are presented in Table 1.

## Overall, match and training incidence

A total of 30 injuries were reported in 15 different players during the three seasons (two match injuries and 28 training injuries) within a total exposure time of 4,446.1 h (310 h of match exposure and 4,136.1 h of training exposure), which is equivalent to an overall incidence rate of 6.75 injuries per 1,000 h of exposure (95% CI [6.47–7.02]). One of the injuries was not taken into account due to the player having to retire from the sport because of the injury. The match injury rate was similar (no statistically ($p > 0.05$) and clinically irrelevant (very likely trivial) differences) to the training injury rate (6.45, 95% CI [6.38–6.52] vs. 6.77], 95% CI [6.50–7.04]/1,000 h) and 38% (15/39) of players sustained at least one injury during the three seasons. Players sustained 0.77 injuries per season on average, which is equivalent to 10 injuries per season for a squad of 13 players.

The injury incidence and characteristics of the injuries during the three seasons are shown in Table 2.

**Table 2 Injury incidence.**

| Injuries | Season 15/16 | | | Season 16/17 | | | Season 17/18 | | | Total | | |
|---|---|---|---|---|---|---|---|---|---|---|---|---|
| | Number (%) | Incidence (95% CI) | Injury burden | Number (%) | Incidence (95% CI) | Injury burden | Number (%) | Incidence (95% CI) | Injury burden | Number (%) | Incidence (95% CI) | Injury burden |
| Overall | 8 | 5.31 [4.9–5.71] | 155.3 | 12 | 90.3 [8.51–9.55] | 108.4 | 10 | 9.09 [8.93–9.25] | 31.7 | 30 | 6.75 [6.47–7.02] | 96.5 |
| Training | 8 (100) | 5.66 [5.26–6.06] | 165.6 | 11 (91.7) | 9.00 [8.50–9.50] | 108.0 | 9 (90) | 6.00 [5.51–6.49] | 30.0 | 28 (93.3) | 6.77 [6.50–7.04] | 99.4 |
| Match | 0 (0) | 0 | 0 | 1 (8.3) | 9.38 [9.22–9.53] | 112.5 | 1 (10) | 6.21 [5.69–6.73] | 54.5 | 2 (6.7) | 6.45 [6.38–6.52] | 58.1 |
| Mechanism | | | | | | | | | | | | |
| Traumatic training | 5 (62.5) | 3.32 [3.01–3.63] | 145.4 | 8 (66.7) | 6.02 [5.59–6.45] | 87.3 | 7 (70) | 4.35 [3.91–4.78] | 16.1 | 20 (66.7) | 4.50 [4.27–4.72] | 82.1 |
| Traumatic match | 0 (0) | 0 | 0 | 1 (8.3) | | | 1 (10) | | | 2 (6.7) | | 2 |
| Overuse training | 3 (37.5) | 1.99 [1.75–2.23] | 10 | 4 (33.3) | 3.01 [2.71–3.31] | 21.1 | 3 (30) | 1.86 [1.58–2.15] | 15.5 | 10 (33.3) | 2.25 [2.09–2.41] | 14.4 |
| Circumstance | | | | | | | | | | | | |
| Contact | 0 (0) | 0 | 0 | 0 (0) | 0 | 0 | 2 (20) | 1.24 [1.01–1.47] | 5.6 | 2 (6.7) | 0.45 [0.38–0.52] | 2 |
| Non-Contact | 8 (100) | 5.31 [4.91–5.71] | 155.3 | 12 (100) | 9.03 [8.51–9.55] | 108.4 | 8 (80) | 4.97 [4.50–5.43] | 26.1 | 28 (93.3) | 6.30 [6.03–6.56] | 94.5 |
| Recurrence | | | | | | | | | | | | |
| No | 8 (100) | 5.31 [4.91–5.71] | 155.3 | 9 (75) | 6.02 [5.59–6.45] | 82.0 | 8 (80) | 4.97 [4.50–5.43] | 25.5 | 25 (83.3) | 5.62 [5.37–5.87] | 86.4 |
| Yes | 0 (0) | 0 | 0 | 3 (25) | 2.26 [2.00–2.52] | 26.3 | 2 (20) | 1.24 [1.01–1.47] | 6.2 | 5 (16.7) | 1.12 [1.01–1.24] | 10.1 |
| Early | 0 (0) | 0 | 0 | 1 (33.3) | 0.75 [0.60–0.90] | 12.0 | 0 (0) | 0 | 0.0 | 1 (20) | 0.22 [0.17–0.28] | 3.6 |
| Late | 0 (0) | 0 | 0 | 2 (66.7) | 1.51 [1.29–1.72] | 14.3 | 1 (50) | 0.62 [0.46–0.78] | 1.9 | 3 (60) | 0.67 [0.59–0.76] | 4.9 |
| Delayed | 0 (0) | 0 | 0 | 0 | 0 | 0 | 1 (50) | 0.62 [0.46–0.78] | 4.3 | 1 (20) | 0.22 [0.17–0.28] | 1.6 |
| Severity | | | | | | | | | | | | |
| 0 days | 0 (0) | 0 | 0 | 0 (0) | 0 | 0 | 0 (0) | 0 | 0 | 0 (0) | 0 | 0 |
| Minimal (1–3 days) | 1 (12.5) | 0.66 [0.52–0.80] | 2.0 | 2 (16.7) | 1.51 [1.29–1.72] | 4.5 | 5 (50) | 3.10 [2.74–3.47] | 6.8 | 8 (26.7) | 1.80 [1.66–1.94] | 4.5 |
| Mild (4–7 days) | 2 (25) | 1.33 [1.13–1.53] | 8.0 | 2 (16.7) | 1.51 [1.29–1.72] | 9.0 | 4 (40) | 2.48 [2.16–2.81] | 13.0 | 8 (26.7) | 1.80 [1.66–1.94] | 10.1 |
| Moderate (8–28 days) | 4 (50) | 2.65 [2.37–2.93] | 51.1 | 7 (58.3) | 5.27 [4.87–5.67] | 94.8 | 6 (60) | 0.62 [0.46–0.78] | 11.8 | 12 (40) | 2.70 [2.52–2.87] | 49.9 |
| Severe (>28 days) | 1 (12.5) | 0.66 [0.52–0.80] | 94.2 | 0 (0) | 0 | 0 | 0 (0) | 0 | 0 | 1 (3.3) | 0.22 [0.17–0.28] | 31.9 |
| Career ending | 0 (0) | 0 | 0 | 1 (8.3) | 0.75 [0.60–0.90] | – | 0 (0) | 0 | 0 | 1 (3.3) | 0.22 [0.17–0.28] | – |
| Position | | | | | | | | | | | | |
| Goalkeeper | 2 (25) | 1.33 [0.40–2.25] | 16.6 | 2 (16.7) | 1.51 [0.12–2.89] | 7.5 | 0 (0) | 0 | 0 | 4 (13.3) | 0.90 [0.34–1.46] | 7.9 |
| Lastwoman | 4 (50) | 2.65 [1.67–3.63] | 40.5 | 7 (58.3) | 5.27 [3.97–6.56] | 77.5 | 4 (40) | 2.48 [1.50–3.46] | 11.2 | 15 (50) | 3.37 [2.74–4.01] | 40.9 |
| Wing | 0 (0) | 0 | 0 | 3 (25) | 2.26 [1.58–2.94] | 23.3 | 6 (60) | 3.73 [2.52–4.93] | 20.5 | 9 (30) | 2.02 [1.57–2.48] | 14.4 |
| Pivot | 2 (25) | 1.33 [0.40–2.02] | 98.2 | 0 (0) | 0 | 0 | 0 (0) | 0 | 0 | 2 (6.7) | 0.45 [0.05–0.85] | 33.3 |

**Note:**
CI, Confidence interval.

## Injury characteristics

### Player position

Lastwomen (3.37, 95% CI [2.74–4.01]/1,000 h) incidence rate was most likely higher (100% likelihood) than wings (2.02, 95% CI [1.57–2.48]/1,000 h), goalkeepers (0.90, 95% CI [0.34–1.46]/1,000 h) and pivots (0.45, 95% CI [0.05–0.85]/1,000 h). Wings had a very likely higher incidence rate (96.6% likelihood) than goalkeepers and most likely higher (100% likelihood) than pivots. Finally, goalkeepers had a likely higher incidence rate (76.6% likelihood) than pivots.

### Injury mechanism

Two out of three injuries were due to trauma and one out of three injuries was due to overuse. The incidence rate of traumatic injuries was most likely higher (100% likelihood) than overuse injuries (4.5, 95% CI [4.27–4.72] vs. 2.25, 95% CI [2.09–2.41]/1,000 h). Most injuries were caused by non-contact situations (93%), with only 7% of injuries occurring during contact situations.

### Injury location

Table 3 shows the injury location and type of injury per season. Lower extremity injuries (5.62 per 1,000 h of exposure, 95% CI [5.37–5.87]) were the most frequently injured location, followed by upper limb injuries (0.67 per 1,000 h of exposure, 95% CI [0.59–0.76]), and then trunk injuries (0.45 per 1,000 h of exposure, 95% CI [0.38–0.52]). No head and neck injuries were reported. The lower extremity region predominantly injured was the thigh (3.37 per 1,000 h of exposure, 95% CI [3.18–3.57]), followed by the ankle (0.90 per 1,000 h of exposure, 95% CI [0.8–1.0]), with the knee, hip/groin and lower leg/Achilles tendon regions demonstrating the same incidence rate (0.45 per 1,000 h of exposure, 95% CI [0.38–0.52]). No foot/toe injuries were reported. In terms of paired-comparisons, thigh injuries occurred more frequently (100% likelihood) than injuries in other lower extremity regions. Ankle injury rates were most likely higher (100% likelihood) than knee, hip/groin and lower leg/Achilles tendon injuries. There were no meaningful differences between the remaining paired combinations.

### Type of injuries

The mean incidence of injury type grouping is presented per 1,000 h of exposure with 95% CIs. Most injuries were diagnosed as muscle/tendon injuries (4.95 per 1,000 h of exposure, 95% CI [4.71–5.18]), followed by joint (non-bone) and ligament (1.35 per 1,000 h of exposure, 95% CI [1.23–1.47]), and fractures and bone stress and contusions with the same injury incidence (0.22 per 1,000 h of exposure, 95% CI [0.17–0.28]). No central/peripheral nervous system injuries and skin lesions were recorded. The most common injury types were hamstring muscle injuries (1.80 per 1,000 h of exposure, 95% CI [0.66–1.94]), followed by quadriceps muscle injuries (1.57 per 1,000 h of exposure, 95% CI [1.44–1.71]), ankle sprains (0.90 per 1,000 h of exposure, 95% CI [0.8–1.0]) and anterior cruciate ligament (ACL) tears (0.45 per 1,000 h of exposure, 95% CI [0.38–0.52]). Muscle/tendon injury incidence rates were most likely higher than other types of injury rates (100%

**Table 3  Injury characteristics and incidence according location and type of injury.**

| Injury location | Season 15/16 | | | Season 16/17 | | | Season 17/18 | | | Total | | |
|---|---|---|---|---|---|---|---|---|---|---|---|---|
| | Number (%) | Incidence (95% CI) | Injury burden | Number (%) | Incidence (95% CI) | Injury burden | Number (%) | Incidence (95%CI) | Injury burden | Number (%) | Incidence (95% CI) | Injury burden |
| Upper limbs | 1 (12.5) | 0.66 [0.52–0.80] | 14.6 | 0 (0) | 0 | 0 | 2 (20) | 1.24 [1.01–1.47] | 2.5 | 3 (10) | 0.67 [0.59–0.76] | 5.8 |
| Shoulder/clavicula | 0 (0) | 0 | 0 | 0 (0) | 0 | 0 | 1 (10) | 0.62 [0.46–0.78] | 0.6 | 1 (3.3) | 0.22 [0.17–0.28] | 0.2 |
| Hand/finger/thumb | 1 (12.5) | 0.66 [0.52–0.80] | 14.6 | 0 (0) | 0 | 0 | 1 (10) | 0.62 [0.46–0.78] | 1.9 | 2 (6.7) | 0.45 [0.38–0.52] | 5.6 |
| Trunk | 1 (12.5) | 0.66 [0.52–0.80] | 2.0 | 0 (0) | 0 | 0 | 1 (10) | 0.62 [0.46–0.78] | 2.5 | 2 (6.7) | 0.45 [0.38–0.52] | 0.7 |
| Lower back/pelvis/sacrum | 1 (12.5) | 0.66 [0.52–0.80] | 2.0 | 0 (0) | 0 | 0 | 1 (10) | 0.62 [0.46–0.78] | 2.5 | 2 (6.7) | 0.45 [0.38–0.52] | 0.7 |
| Lower limbs | 6 (75) | 3.98 [3.64–4.33] | 138.7 | 12 (100) | 9.03 [8.51–9.55] | 108.4 | 7 (70) | 4.35 [3.91–4.78] | 26.7 | 25 (83.3) | 5.62 [5.37–5.87] | 90.0 |
| Hip/groin/adductor | 0 (0) | 0 | 0 | 1 (8.3) | 0.75 [0.60–0.90] | 16.6 | 1 (10) | 0.62 [0.46–0.78] | 2.5 | 2 (6.7) | 0.45 [0.38–0.52] | 5.8 |
| Thigh | 3 (37.5) | 1.99 [1.75–2.23] | 22.6 | 7 (58.3) | 5.27 [4.87–5.67] | 48.9 | 5 (50) | 3.10 [2.74–3.47] | 20.5 | 15 (50) | 3.37 [3.18–3.57] | 29.7 |
| Hamstrings | 1 (12.5) | 0.66 [0.52–0.80] | 10.0 | 5 (41.7) | 3.76 [3.43–4.10] | 33.1 | 2 (20) | 1.24 [1.01–1.47] | 3.1 | 8 (26.7) | 1.80 [1.66–1.94] | 14.4 |
| Quadriceps | 2 (25) | 1.33 [1.13–1.53] | 12.6 | 2 (16.7) | 1.51 [1.29–1.72] | 15.8 | 3 (30) | 1.86 [1.58–2.15] | 17.4 | 7 (23.3) | 1.57 [1.44–1.71] | 15.3 |
| Knee | 1 (12.5) | 0.66 [0.52–0.80] | 94.2 | 1 (8.3) | 0.75 [0.60–0.90] | 0 | 0 (0) | 0.0 | 0 | 2 (6.7) | 0.45 [0.38–0.52] | 31.9 |
| Lower leg/Achilles tendon | 1 (12.5) | 0.66 [0.52–0.80] | 4.0 | 1 (8.3) | 0.75 [0.60–0.90] | 9.0 | 0 (0) | 0.0 | 0 | 2 (6.7) | 0.45 [0.38–0.52] | 4.0 |
| Ankle | 1 (12.5) | 0.66 [0.52–0.80] | 17.9 | 2 (16.7) | 1.51 [1.29–1.72] | 33.9 | 1 (10) | 0.62 [0.46–0.78] | 3.7 | 4 (13.3) | 0.90 [0.80–1.00] | 17.5 |
| **Injury type** | | | | | | | | | | | | |
| Fracture and bone stress | 0 | 0 | 0 | 0 | 0 | 0 | 1 (10) | 0.62 [0.46–0.78] | 1.9 | 1 (3.3) | 0.22 [0.17–0.28] | 0.7 |
| Fracture | 0 | 0 | 0 | 0 | 0 | 0 | 1 (10) | 0.62 [0.46–0.78] | 1.9 | 1 (3.3) | 0.22 [0.17–0.28] | 0.7 |
| Joint (non-bone) and ligament | 3 (37.5) | 1.99 [1.75–2.23] | 126.8 | 2 (16.7) | 1.51 [1.29–1.72] | 21.1 | 1 (10) | 0.62 [0.46–0.78] | 0.6 | 6 (20) | 1.35 [1.23–1.47] | 49.5 |
| Sprain/Ligament injury | 3 (37.5) | 1.99 [1.75–2.23] | 126.8 | 2 (16.7) | 1.51 [1.29–1.72] | 21.1 | 1 (10) | 0.62 [0.46–0.78] | 0.6 | 6 (20) | 1.35 [1.23–1.47] | 49.5 |
| Muscle and tendon | 5 (62.5) | 3.32 [3.01–3.63] | 28.5 | 10 (83.3) | 7.53 [7.05–8] | 87.3 | 7 (70) | 4.35 [3.91–4.78] | 25.5 | 22 (73.4) | 4.95 [4.71–5.18] | 45.0 |
| Muscle rupture/tear/strain/cramps | 4 (50) | 2.65 [2.37–2.93] | 24.6 | 9 (75) | 6.77 [6.32–7.23] | 74.5 | 7 (70) | 4.35 [3.91–4.78] | 25.5 | 20 (66.7) | 4.5 [4.27–4.72] | 39.8 |
| Tendon injury/rupture/tendinosis/bursitis | 1 (12.5) | 0.66 [0.52–0.80] | 4 | 1 (8.3) | 0.75 [0.60–0.90] | 12.8 | 0 | 0 | 0 | 2 (6.7) | 0.45 [0.38–0.52] | 5.2 |

**Note:**
CI, Confidence interval.

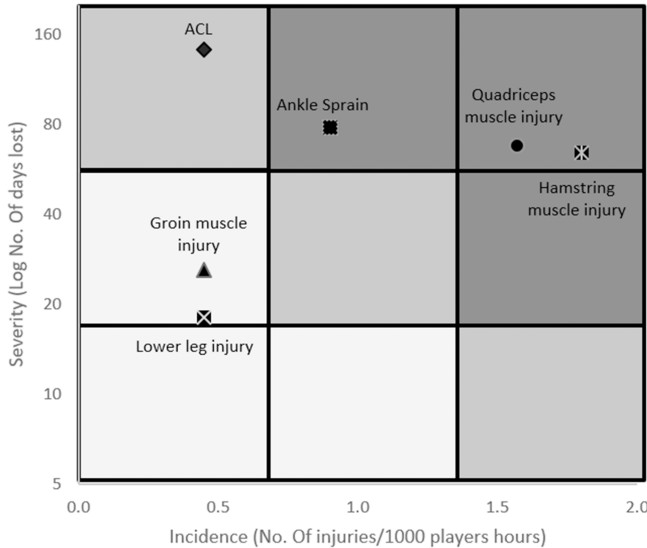

**Figure 1** Quantitative risk matrix of injuries, illustrating the relationship between the severity (consequence) and incidence (likelihood) of the most common injuries.

likelihood). Likewise, joint (non-bone) and ligament incidence rate were most likely higher (100% likelihood) than fractures, bone stress and contusions.

## Severity of injuries

Concerning the severity of injuries, moderate injuries (2.70 per 1,000 h of exposure, 95% CI [2.52–2.87]) were the most usual injuries, followed by minimal and mild injuries (1.80 per 1,000 h of exposure, 95% CI [1.66–1.94]), and finally severe and career ending injuries (0.22 per 1,000 h of exposure, 95% CI [0.17–0.28]). No 0 days injuries were recorded.

Comparisons between each severity level showed that the moderate injury incidence rates were most likely higher (100% likelihood) than other severities. Minimal and mild injury incidence rates were most likely higher (100% likelihood) than severe and career ending injuries.

The recorded overall time-loss injuries was 429 days, so overall injury burden during the three seasons was 96.5 days loss/1,000 h exposure (58.1 in matches and 99.4 in trainings). Figure 1 shows a quantitative risk matrix illustrating the relationship between the severity and incidence of the most common reported injuries. For each injury type, severity is shown as the average number of days lost (log scale), while incidence is shown as the number of injuries per 1,000 h of total exposure for each injury type. The shading illustrates relative importance of each of the injury types; the darker the color, the greater the injury burden, and the greater the priority should be given to prevention. Furthermore, lastwomen and pivots showed the highest injury burden (40.9 and 33.3 days loss/1,000 h exposure) compared to goalkeepers and wings (7.9 and 14.4 days loss/1,000 h exposure). On the other hand, muscle/tendon injuries and joint (non-bone) and ligament injuries showed similar injury burden (44.98 and 49.48 days loss/1,000 h exposure) although their overall incidence was significantly different. Regarding injury

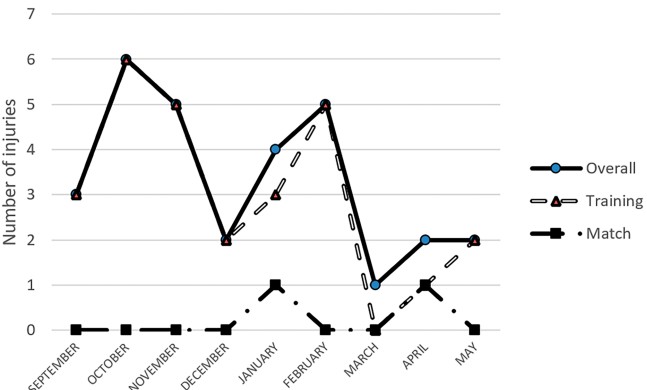

**Figure 2  Distribution of total injury incidence.**     

location, the knee showed a significantly higher injury burden (31.9 days loss/1,000 h exposure) compared to the rest of the lower extremity muscle groups (ankle: 17.5; quadriceps: 15.3; hamstring: 14.4; hip/groin: 5.8 and lower leg/Achilles tendon: 4.0).

### Recurrent injuries

The incidence rate of new injuries (5.62 per 1,000 h of exposure, 95% CI [5.37–5.87]) was most likely higher (100% likelihood) than recurrent injuries incidence rate (1.12 per 1,000 h of exposure, 95% CI [1.01–1.24]). One-fifth of the overall injuries were recurrent injuries; of these, 20% of injuries were classified as "early recurrence" (within 0–2 months); 60% of injuries were classified as "late recurrence" (2–12 months); and 20% of injuries were classified as "delayed recurrence" (>12 months) (*Fuller et al., 2006*). The most common recurrent injury was quadriceps and hamstring strains. Regarding injury burden, new injuries had a significantly higher injury burden compared to recurrent injuries (86.4 vs. 1.1 days lost/1,000 h exposure).

### Season variation of injury pattern

Figure 2 illustrates monthly distribution of injuries, both overall, during training and match over the three seasons. The highest incidence of injuries was observed in October (1.35 per 1,000 h of exposure, 95% CI [1.23–1.47]). Training and match number of injuries follow a similar trend, in which the risk of injuries was higher in the early stages of the season and post winter/Christmas break.

## DISCUSSION

The overall, training and match incidence rates reported in the current study were comparable to those found in the only study (to the authors knowledge) that has provided three incidence rates separately in a cohort of 17 female futsal players (*Angoorani et al., 2014*) (4.7, 3.1 and 10.7 injuries per 1,000 h of exposure to overall, training and match play, respectively). Conversely, the match injury incidence reported in the current study (6.4 injuries per 1,000 h of match play) is lower than that reported by *Hamid, Jaafar & Ali (2014)* in the Malaysian female futsal league (29.6 injuries per 1,000 h of exposure to match play). An explanation of this discrepancy may be attributed to the more congested

competitive calendar in the study carried out by *Hamid, Jaafar & Ali (2014)* compared to our study. Thus, while in their study the Malaysian league had a duration of approximately 22 weeks (1st July until 28th November) with a break in August (because of fasting during Ramadan) and one or two matches per week, the three seasons (2015–2018) of the Spanish second division analyzed in the current study lasted 8 months (average of 34.3 ± 2.1 weeks) with two breaks periods of 2–3 weeks (at Christmas and Easter) with one match played per week (usually at the weekend days). This hypothesis may be supported by evidence from prospective epidemiological studies carried out in elite male futsal players (*Ribeiro, Oliveira & Costa, 2006*; *Junge & Dvorak, 2010*) and football players (*Dvorak et al., 2011*; *Junge & Dvořák, 2015*) during international tournaments (i.e., World cups) which have shown higher incidence rates in comparison with those conducted during national league futsal (*Hamid, Jaafar & Ali, 2014*; *Álvarez Medina et al., 2016*) and football (*Noya Salces et al., 2014*; *Stubbe et al., 2015*). This is likely due to the higher match demands during international tournaments with relatively shorter recovery times between matches. These tournaments also tend to occur at the end of long competitive league seasons where accumulated fatigue may also be a factor in the higher incidence rates.

Unlike data from other team sports (regardless of the sex of the players) (i.e., football (*Giza et al., 2005*; *Waldén, Hägglund & Ekstrand, 2007*), basketball (*Borowski et al., 2008*), netball (*Best, 2017*)) where match injury incidence is always notably higher (almost 10 times) than the injury rate obtained for training sessions, in our study both incidence rates were similar. The latest trends in strength and conditioning for team sports have suggested that training session design (i.e., work-load, intensity, duration), when possible, should mimic match demands so that players are better prepared for what they face during matches (*Gabbett, 2016*). Perhaps, the training sessions designed by the team staff might have included a large number of repeated high-intensity actions (e.g., accelerations and decelerations, changes of direction) in order to replicate the evolving nature of the futsal game. However, an excessive training load and/or an insufficient recovery of previous efforts might have forced players to perform some of these highly demanding training sessions under suboptimal states of readiness and this could have potentially increased the risk of injuries (mainly muscle-tendon and ligament injuries) (*Gabbett, 2004*). To determine whether or not futsal players are in an optimal state of readiness for the stress that will be a priori elicited by training, it is advisable to monitor daily training load (internal and external) and strain, wellbeing and recovery status from previous efforts and also include regular physical performance tests as a component of the training program (*Bouaziz et al., 2016*; *Elloumi et al., 2012*). This information might help coaches and physical trainers to constantly re-adjust the design of the training sessions throughout the season so that the physical and psychological demands that will be imposed on the players do not negatively affect their optimal readiness to re-perform.

When exploring differences in playing position on incidence rates our data from the goalkeepers and outfield player's differed from the findings previously reported by *Hamid, Jaafar & Ali (2014)*. Their study, also in female futsal players, showed a higher incidence rate in goalkeepers but we found outfield players showed higher incidence and higher amount of days off per injury than goalkeepers. Our findings are similar to that which has

been reported in other team sports such as handball (*Tsigilis & Hatzimanouil, 2005*) and football (*Mallo et al., 2011*; *Falese, Della Valle & Federico, 2016*). It is difficult to prescribe a reason for the discrepancy between the findings of *Hamid, Jaafar & Ali (2014)* and our current study. However, it might be due to the fact that outfield players need to perform a larger number of repeated high intensity multiplanar movements that occur every few seconds (*Doğramacı & Watsford, 2006*), which may place outfield players at a higher risk of injury than goalkeepers.

Previous studies have indicated that a large percentage of injuries in male futsal players (*Ribeiro, Oliveira & Costa, 2006*; *Junge & Dvorak, 2010*) are caused by contact trauma, however the current study demonstrates that most injuries sustained by female players are due to non-contact trauma (>90%). Our results are in agreement with the study of *Angoorani's et al. (2014)* and might be partly attributed to the fact that both studies included training injury incidence data, something that other studies have failed to do. Furthermore, the higher number of high intensity phases observed in elite male players during the course of futsal play (*Carling et al., 2015*; *Naser, Ali & Macadam, 2017*) might contribute to generate more tackling situations and partially explain the fact that males suffer more contact injuries than females.

With respect to the location of futsal-related injuries, and similar to previous studies in male (*Ribeiro, Oliveira & Costa, 2006*; *Junge & Dvorak, 2010*; *Álvarez Medina et al., 2016*) and female futsal players (*Angoorani et al., 2014*; *Hamid, Jaafar & Ali, 2014*), lower extremity injuries were, by far, the most frequent injuries (83.3% of all the injuries recorded). The thigh (50% of all the injuries recorded) was the anatomical region of the lower extremity where injuries occurred significantly more followed by the knee (6.7% of all the injuries recorded) and ankle (6.7% of all the injuries recorded). Furthermore, the most common type of injury grouping was muscle/tendon injuries followed by joint (non-bone) and ligament injuries. As futsal is a fast-paced game relying mostly on the lower extremity for ball control, involving sprinting and frequent changes in direction such observations were anticipated. In football, it has been demonstrated that player match availability has a strong correlation ($r > 0.85$) with team success (i.e., ranking position, games won, goals scored, total points) (*Eirale et al., 2013*; *Hägglund et al., 2013*; *Carling et al., 2015*). If this statement also holds for futsal, then injury prevention measures should focus not just on reduction of the incidence of the most frequent injuries but also on reduction of the injuries with the highest burden (e.g., those injuries that keep players out of training and match play the longest) (*Bahr, Clarsen & Ekstrand, 2017*). According to the results found in this study, knee and thigh injuries are those with the highest injury burden with 31.9 and 29.7 days of absence per 1,000 player hours, respectively. In particular, medical and fitness staff should implement measures mainly aimed (but not solely) at reducing the number and severity of ACL and hamstring and quadriceps muscle injuries. It should be noted that one player from the team had to retire from futsal due to an ACL rupture, which was not included in the injury burden calculation as the number of days lost were not defined. This reinforces the need to deliver targeted interventions aimed at reducing this devasting and relatively frequent (two cases in the three seasons recorded in our study for a single team) type of injury in female athletes. It should be also

highlighted that the overall (31.7 days) and training (30 days) injury burdens of the last season analyzed (2017/18) were significantly lower than those obtained for the two previous seasons (overall = 155.3 (2015/16) and 108.4 (2016/17) days; training = 165.6 (2015/16) and 108.0 (2016/17) days). Perhaps, the fact that during the three seasons that were object of study the club kept the same medical staff and head coach may have been a factor that may explain in part this circumstance. In this sense, and similar to what was found in previous studies (*Ekstrand et al., 2018*; *Lausic et al., 2009*), the potential and gradual improvement in the quality of the internal communication not only within the members of the medical staff but also between the medical staff and the coach that might have occurred throughout the three consecutive seasons may have had a positive impact on the players' availability for futsal play in the last season. In fact, according to *Ekstrand et al. (2018)*, the measures designed to reduce the injury burden in elite teams should not only address the traditionally proposed modifiable injury risk factors, for example, eccentric strength deficits (*Croisier et al., 2008*; *Petersen et al., 2011*; *Van Dyk et al., 2016*), poor neuromuscular control (*Lees & Nolan, 1998*; *Hewett et al., 2005*), altered muscle architecture (*Lees & Nolan, 1998*; *Arnason et al., 2004*; *Timmins et al., 2016*), player load and match frequency (*Rahnama et al., 2003*; *Miloski, Freitas & Barra-Filho, 2012*) but also some new external factors such as job security and club stability and players adherence and coaches compliance to the injury prevention programs applied.

The inclusion of updated and evidence-based advancements in factors related to injury management (including diagnosis techniques, treatment approaches and monitoring tools) might also have a positive impact on the injury burden.

As expected, new injury rates were higher than recurrent injury incidence rates (5.6 vs. 1.1 injuries per 1,000 h). However, the recurrent rate identified in the present study may be considered high. It was found that 20% of recurrent injuries (mainly lower extremity muscle and tendon injuries) occurred within 2 months after return to play. This may be regarded as a sign of premature return to train/play and incomplete or inadequate rehabilitation. The lack of and evidence-based criteria for a safe return to train/play may have resulted in letting injured players return to play sooner than recommended. This may have been due to the desire to let them play in important matches or to let them play with ongoing minor symptoms, and this might be two primary reasons behind the high recurrent injury incidence rate. Future studies should extend our current knowledge further in relation to the improvement of the decision-making process for a safe return to train/play by developing learning algorithms or artificial intelligence-based models that allow the identification of when a player is successfully rehabilitated before returning to train/play. Furthermore, medical and fitness team staff should allow players enough time for rehabilitation before return to train/play.

Regarding the moment when most injuries took place, the findings indicate that there are two periods when they are more likely to occur, October and January–February. The higher amount of injuries during October may be explained by the fact that within the pre-season period the training loads are much higher than during the competitive period (*Miloski, Freitas & Barra-Filho, 2012*) and accumulating fatigue may increase the injury

risk during the first weeks of competition. *Petersen et al. (2010)* reported a higher incidence in the 2 months after the winter break (January–February) which is consistent with the results of the present study.

### Limitations

Despite being one of the first prospective studies that has analyzed the incidence rates and characteristics of futsal related injuries in female players, some limitations must be considered. The sample size of players and injuries is small, and results should be cautiously interpreted (especially the incidence rates reported for specific and less frequent injuries). The analysis of only one team limits the external validity of the results. Consequently, it is unknown if female players from other teams in which there could be a higher (or lower) medical staff-to-player ratio or access to other staff (such as strength and conditioning coaches, psychologists and nutritionists) may show similar injury incidence rates and characteristics than those reported in the current study. Even though all female players had sub-elite status, most of them had jobs besides futsal that could alter their risk of injury and recovery time, for example, by preventing them from training or taking full advantage of medical treatment. Therefore, future studies are needed in order to analyze if elite female futsal players on full-time (professional) contracts may show different injury incidence rates, characteristics and burden.

## CONCLUSIONS

Sub-elite female futsal players (particularly outfield players) are exposed to a substantial risk of sustaining injuries. Most injuries had a non-contact mechanism, with the lower extremity the most frequently injured anatomical region. Knee (ACL tears) and thigh (hamstring and quadriceps muscle strains) injuries are those with the highest injury burden. Special attention should be given to the first weeks of competition after pre-season and soon after the Christmas break as incidence rates peak during this period in female futsal players. Medical and fitness team staff should focus their attention on designing, implementing and then evaluating preventative measures that target the most common diagnoses, namely, ligament and muscle/tendon injuries highlighted in this study, as well as making sure that return to train/play criteria are in place in order to reduce the injury burden within female sub-elite futsal players.

### Funding

Iñaki Ruiz-Pérez was supported by a pre-doctoral grant from the Ministerio de Economía y Competitividad (FPI BES-2015-07200) from Spain. Francisco Ayala was supported by a postdoctoral grant from Seneca Foundation (postdoctoral fellowships funded by the regional sub program focuses on the postdoctoral development, 20366/PD/17) from Spain. The funders had no role in study design, data collection and analysis, decision to publish, or preparation of the manuscript.

## Grant Disclosures

The following grant information was disclosed by the authors:
Ministerio de Economía y Competitividad: FPI BES-2015-07200.
Seneca Foundation (postdoctoral fellowships funded by the regional sub program focuses on the postdoctoral development): 20366/PD/17.

## Competing Interests

The authors declare that they have no competing interests.

## Author Contributions

- Iñaki Ruiz-Pérez conceived and designed the experiments, prepared figures and/or tables, authored or reviewed drafts of the paper, approved the final draft.
- Alejandro López-Valenciano conceived and designed the experiments, analyzed the data, prepared figures and/or tables, authored or reviewed drafts of the paper, approved the final draft.
- Alejandro Jiménez-Loaisa conceived and designed the experiments, performed the experiments, authored or reviewed drafts of the paper, approved the final draft.
- Jose L.L. Elvira conceived and designed the experiments, contributed reagents/materials/analysis tools, authored or reviewed drafts of the paper, approved the final draft.
- Mark De Ste Croix conceived and designed the experiments, authored or reviewed drafts of the paper, approved the final draft.
- Francisco Ayala conceived and designed the experiments, authored or reviewed drafts of the paper, approved the final draft.

## Human Ethics

The following information was supplied relating to ethical approvals (i.e., approving body and any reference numbers):

The University Office for Research Ethics (Órgano evaluador de proyectos, Universidad Miguel Hernández de Elche) approved the study (DPS.FAR.02.14).

## Data Availability

The raw measurements are available in the Supplemental Files.

## Supplemental Information

Supplemental information for this article can be found online at http://dx.doi.org/10.7717/peerj.7989#supplemental-information.

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
