# Peer review of "Injury incidence, characteristics and burden among female sub-elite futsal players: a prospective study with three-year follow-up"

_PeerJ, doi:10.7717/peerj.7989_

## Round 0.1 · original submission · Major Revisions

Your manuscript has been reviewed by several reviewers and found to require multiple revisions before further consideration. Please provide a point by point response regarding how in the revised submission you addressed each issue raised by each reviewer.

Reviewer 1 ·

Basic reporting

The English language is clear.
The introduction was able to present the study problematic.
The paper structure follow partially the conforms to journal guidelines.

Experimental design

Original research within journal scope.
The research question is relevant for female elite futsal players.
The investigation present ethical standard.
The Methods section (procedures) is well described with sufficient detail to replicate the study. However, participants division is not clear.

Validity of the findings

The results are not reliable for female elite futsal players.
Statistic well done.
Conclusion well stated.
Many speculations were supplied.

Additional comments

The manuscript entitled “Injury incidence, characteristics and burden among female elite futsal players: A prospective study with three-year follow-up” aimed to identify injury incidence, characteristics and burden among elite female futsal players. This topic is very interesting, especially for female players. However, the objective of study is not according with methods presented. The study was prospective with three follow-up of a specific futsal team. I have some concerns that are listed below.

Major concerns
The authors stated that 39 players were volunteered in the study, and they were prospectively followed during three consecutive seasons. According with results 14, 13 and 12 players were prospectively followed during one year. Thus, the title did not correspond to objective of study. Questions: Are different players in each season? Are the same players in each season?

How the authors define elite players? I am not sure about result reliability for elite players, since the study assessed just one team of second division. If the authors had followed more teams of second division, the results would be reliable to show injury incidence and burden among female Spanish futsal players (second division).
Considering these major concerns, the results of study are not reliable to show the incidence, characteristics and burden of female elite futsal players. The results are reliable only for one specific team.

Specific comments
Introduction
Ok.

Material and Methods
Line 100: The sample size and players level (second division) is the weakness of the study. In the discussion section, I suggest the authors present these limitations.
Lines 100-101: How many players participate of the study? In the table 1, the authors showed that team size was 14, 13 and 12 players in the seasons 15/16, 16/17 and 17/18, respectively.
Line 158: The authors’ statement that player match hour were calculated by 40 min with stopped clock/60. Is this equation indicates match hour for each player? Since match participation is different for each players during the season.

Results
Table 2: In the column “Season 16/17 Incidence (95%CI)”, in the first line there is a mistyping. I believe that the incidence is 9.03 and not 90.3.
Line 227: The authors statement “Lower extremity injuries (5.62 per 1000 hours of exposure) were the most frequently injured location.” I cannot see this information in the table 3 (line with this information seems belong to trunk). The table is confusing. Please adjust table 3.
Lines 266-270: These information can be represented in table (Suggestion).

Discussion
Lines 334-338: As suggestion, following the results section, in this paragraph the discussion is about mechanism and severity of injury.
Line 411: Change “simple” for “sample”.
Lines 412-413: I am not sure about the elite status of players. Since they participated of Spanish second division, and perhaps the injury incidence can be different for first division players. Furthermore, the study assessed only one team.

Conclusion
Lines 419-420: Table 3 shown that ankle strain had similar injury burden in comparison to hamstring and quadriceps

Reviewer 2 ·

Basic reporting

The manuscript is well structured and well written. I suggest minor corrections that also follow in the word document (identified with track-changes).

Experimental design

The experimental design is clear. I suggest some minor alterations in terminology.

Validity of the findings

Findings seem valid, however as stated in the limitations, the number of athletes is very low for this kind of studies.

Additional comments

I congratulate the authors on the theme of the manuscript and on the effort to follow the team for 3 seasons. The study shows a carefully thought-out research setup. I suggest a minor revision.
Please consider the following comments:
1. As the athletes belong the the second division, I do not consider the to be elite. I suggest sub-elite.
2. I considerer misleading referring that 39 athletes from the same team were prospectively followed during three consecutive seasons (2015-2018). It is not true. Please make the adequate adjustment to the text.
3. lines 35-40 do not link adequately the the rest of the abstract. Please make the transition smoother.
4. lines 143-144. I do not agree on the arrangement made to analyse severity of injuries. You do not follow the recommendations of Fuller (2006) although you refer the author. Please make the necessary changes to be in agreement with Fuller (2006). I consider that 0 days (with time-loss) are separate from slight injuries (1-3 days). I would consider in this arrangement the use of (career ending injury) as you have a career ending injury in the study.
5. line 164. What study did you use to make the statistical power calculations? Please make the reference.
6. line 262 and others. I recommend the use of time-loss injuries instead of "time lost".
7. 281-283 please make the reference to the authors/study.

Annotated reviews are not available for download in order to protect the identity of reviewers who chose to remain anonymous.

·

Basic reporting

Authors have successfully highlighted gaps on the current knowledge to support the study.

For the most part of the manuscript, it was written in a clear and concise manner. However some errors in spelling/typos were noted.
1. Line 369 (e.g de longest)
2. Line 411 ."The simple size ...."

Experimental design

In the data collection section (2.2)
1. It is not yet clear on the flow of assessment of injured players:
a. What happened to them when they sustained injury?
i. At training and during competition - (assess and treated on the side line)
ii. Who was/were responsible to assess and diagnosed the injured player and decide
on the status of the player following injury?
iii. Were diagnoses based on clinical judgements only? Were other forms of
assessments performed?
2. Line 127 - 128. "Previous injuries were not taken into consideration,..... selection". The
sentence is unclear. What did you mean by not taken into consideration?
3. Line 128 - 129. "Those individuals ... established after discussion were included, ..
medical staff."
i. Did you mean included into the statistical analyses?
ii. Does "recovery period was established" referring to estimated duration of
recovery?
4. Line 130 - 131. " Previous injuries of those ... were not included." The sentence is
unclear.
i. However author stated that injuries were also classified into new and recurrent
injuries. Did the author only refers to recurrence of injuries within the 3 seasons of
study period? Please clarify.
5. Line 145. "Illnesses, any physical or mental complaint .... were excluded". What about
any condition (illnesses, any physical or mental compliant that resulted in time away
from activities?
6. Line 148. Are you referring stature to height? If so suggests to use the term height as
that would be clearer.

Validity of the findings

Discussion section
2nd paragraph: Line 320 - 333
1. The current study found no difference in the incidence of injuries during match vs
training, which was contrary to those reported by earlier researchers (refer to line 320
322). Suggest authors to discuss potential factors that could be responsible for such
observation.
4th Paragraph:
2. While instilling a preventative injury program might have an impact on reducing injury
burden to a certain extent. Factors related to injury management including advancement
in treatment approach could also affect speed of recovery.

Additional comments

Thank you for giving me the opportunity to review this manuscript. I felt that the manuscript explores area that of interest particularly in the field of Sports Medicine.

Reviewer 4 ·

Basic reporting

Congratulation to the authors for the study performed. This is a well-written manuscript that presents interesting results based on rational working hypothesis. However, some points should be clarified before publication.

Experimental design

The objectives are concise and agree with the theoretical framework proposed in the Introduction section. The logic of this study is clear.

Validity of the findings

The methodological questions are clear, and the results obtained are in accordance with the objectives and hypotheses raised.

Additional comments

However, I offer the following suggestions for improving the quality of the document:
1. The introduction provides a good perspective of the main topic, However to make the introduction more substantial, the author may wish to provide several actual references
2. Explain in more detail the participants of each season.
3. Explain training application of the results obtained for female futsal players
4. Explain the limitations of the study

---

## Round 0.2 · Minor Revisions

Although the manuscript was substantially revised, there are still a few minor revisions that need to be made. Please ensure the corrections suggested by all reviewers including those made in the word document (identified with track-changes) are assessed.

Reviewer 2 ·

Basic reporting

I reinforce that the manuscript is well structured and well written. I suggest minor corrections that also follow in the word document (identified with track-changes).

Experimental design

The experimental design is clear. I suggest some minor alterations in terminology.

Validity of the findings

Findings seem valid, however the low incidence of injuries in the last season do need some explanation.

Additional comments

Thank you for considering my suggestions.


Please consider the following comments:

1.
Line 63: Might want to change from elite to any kind of level futsal practice

2.
Line 110: players analyzed?

3.
Line 137. Please refer to ultrasound imaging.

4.
Lines 203 and 2014. Why?

5.
Line 211: 2007 not 2017.

6.
As stated in the first revision “lines 243-244. I do not agree on the arrangement made to analyze the severity of injuries. You do not follow the recommendations of Fuller (2006) although you refer the author. Please make the necessary changes to be in agreement with Fuller (2006). I consider that 0 days (with time-loss) are separate from slight injuries (1-3 days). I would consider in this arrangement the use of (career ending injury) as you have a career ending injury in the study.”
Please rephrase lines 243-244 to take in consideration fullers 2006 injury classification. Was it accounted for or not? It is not clear to me.

7.
Also, Change TABLE 2 to “career ending”

8.
Please explain the low incidence of injuries in the last season analyzed.

Annotated reviews are not available for download in order to protect the identity of reviewers who chose to remain anonymous.

·

Basic reporting

Author have addressed all the comments made by the reviewer in the current manuscript (revised version).

Experimental design

Author have addressed all the comments made by the reviewer in the current manuscript (revised version).

Validity of the findings

Author have addressed all the comments made by the reviewer in the current manuscript (revised version).

Additional comments

Author have addressed all the comments made by the reviewer in the current manuscript (revised version).

Reviewer 4 ·

Basic reporting

-

Experimental design

-

Validity of the findings

-

Additional comments

The manuscript provides very interesting information, and I believe the results and conclusions will be very useful for researchers in this topic. I really appreciate the authors have followed the suggestions made and I consider that the manuscript is ready to be published in this journal.

---

## Round 0.3 · accepted · Accept

The manuscript has been sufficiently revised.